# The Impact of Antibiotic Therapy on Intestinal Microbiota: Dysbiosis, Antibiotic Resistance, and Restoration Strategies

**DOI:** 10.3390/antibiotics14040371

**Published:** 2025-04-03

**Authors:** Gaia Cusumano, Giancarlo Angeles Flores, Roberto Venanzoni, Paola Angelini

**Affiliations:** 1Department of Chemistry, Biology and Biotechnology, University of Perugia, Via del Giochetto, 06122 Perugia, Italy; gaia.cusumano@dottorandi.unipg.it (G.C.); giancarlo.angelesflores@unipg.it (G.A.F.); roberto.venanzoni@unipg.it (R.V.); 2Centro di Ricerca per l’Innovazione, Digitalizzazione, Valorizzazione e Fruizione del Patrimonio Culturale e Ambientale (CE.D.I.PA.), Piazza San Gabriele dell’Addolorata, 4, 06049 Spoleto, Italy

**Keywords:** antibiotic resistance, dysbiosis, fecal microbiota transplantation, gut microbiota, microbial diversity, probiotics, prebiotics and synbiotics

## Abstract

The human gut microbiota—an intricate and dynamic ecosystem—plays a pivotal role in metabolic regulation, immune modulation, and the maintenance of intestinal barrier integrity. Although antibiotic therapy is indispensable for managing bacterial infections, it profoundly disrupts gut microbial communities. Such dysbiosis is typified by diminished diversity and shifts in community structure, especially among beneficial bacterial genera (e.g., *Bifidobacterium* and *Eubacterium*), and fosters antibiotic-resistant strains and the horizontal transfer of resistance genes. These alterations compromise colonization resistance, increase intestinal permeability, and amplify susceptibility to opportunistic pathogens like *Clostridioides difficile*. Beyond gastrointestinal disorders, emerging evidence associates dysbiosis with systemic conditions, including chronic inflammation, metabolic syndrome, and neurodegenerative diseases, underscoring the relevance of the microbiota–gut–brain axis. The recovery of pre-existing gut communities post-antibiotic therapy is highly variable, influenced by drug spectrum, dosage, and treatment duration. Innovative interventions—such as fecal microbiota transplantation (FMT), probiotics, synbiotics, and precision microbiome therapeutics—have shown promise in counteracting dysbiosis and mitigating its adverse effects. These therapies align closely with antibiotic stewardship programs aimed at minimizing unnecessary antibiotic use to preserve microbial diversity and curtail the spread of multidrug-resistant organisms. This review emphasizes the pressing need for microbiota-centered strategies to optimize antibiotic administration, promote long-term health resilience, and alleviate the disease burden associated with antibiotic-induced dysbiosis.

## 1. Introduction

The human gut microbiota—a complex consortium of trillions of microorganisms—exerts fundamental roles in metabolism, immunomodulation, and pathogen defense. In addition, it influences the enteric nervous system (ENS), a dense network of neurons and glial cells governing gut motility, secretion, absorption, immune responses, and intestinal permeability [1]. Thus, perturbations in microbial communities (dysbiosis) correlate not only with gastrointestinal but also systemic pathologies [2]. Dysbiosis arises from environmental perturbations, immune imbalances, or antibiotics [3]. While antibiotics play a vital role in treating bacterial infections, they can drastically diminish gut microbial diversity, undermining metabolic functions [3,4,5]. This breakdown in microbial homeostasis compromises colonization resistance and facilitates the horizontal gene transfer of antibiotic resistance genes (ARGs), complicating infection control [6,7,8]. Broad-spectrum antibiotics (e.g., β-lactams, fluoroquinolones) are especially disruptive, eradicating key beneficial taxa (e.g., *Bifidobacterium*, *Eubacterium*) and promoting the expansion of multidrug-resistant organisms, such as vancomycin-resistant enterococci [2,8,9]. Dysbiosis-related changes extend to extra-intestinal conditions, including metabolic syndrome, chronic inflammation, and neurodegenerative disorders, underlining the microbiota–gut–brain axis’ significance [10]. This axis mediates neuroimmune and neuroendocrine signaling, affecting both the central nervous system (CNS) and ENS [1,11,12]. Alterations in critical genera (e.g., *Akkermansia*, *Clostridium*) can impair gut motility and reduce the expression of neuronal markers crucial to ENS stability [2,13]. Moreover, antibiotic-induced dysbiosis intensifies gut disorders by increasing intestinal permeability, delaying transit time, and compromising ENS neuronal integrity [1,2,14]. Clinically, these dysregulations heighten infection risks (e.g., *C. difficile*) and bloodstream infections, particularly in critically ill individuals [3,15]. The intestinal barrier—maintained by epithelial and immune cells—becomes more permeable under dysbiosis [16], wherein specialized epithelial cells (Paneth, goblet, enteroendocrine) secrete antimicrobial peptides and sustain mucosal integrity [17].

Multiple therapeutic strategies have emerged to counter dysbiosis. Fecal microbiota transplantation (FMT), probiotics, and synbiotics have proven effective in restoring gut microbial diversity and resilience [3]. Concurrently, antibiotic stewardship programs strive to restrain unnecessary use, maintaining beneficial taxa and diminishing selective pressures driving ARGs [8]. Recent evidence highlights the essential interplay between intestinal epithelial and immune cells in sustaining microbial homeostasis [9,17,18,19]. Ultimately, the gut microbiota underpins an array of physiologic and pathophysiologic processes. This review elucidates the mechanisms behind antibiotic-induced dysbiosis—including microbial imbalances, the emergence of resistance, and compromised host–microbiome interactions—and advocates integrating microbiota-focused strategies into clinical practice to optimize antibiotic usage, encourage robust health, and lessen disease risk.

## 2. Material and Methods

This review systematically examines the effects of antibiotics on the composition and function of the gut microbiota by conducting a comprehensive analysis of the existing literature. Studies published between 2012 and 2025 were identified through searches in PubMed, Scopus, and Web of Science using keywords such as “gut microbiota”, “dysbiosis”, “antibiotic resistance”, and “microbial diversity”. Inclusion criteria prioritized studies reporting both quantitative and qualitative changes in microbial communities following antibiotic exposure. Particular attention was given to broad-spectrum antibiotics—especially β-lactams and fluoroquinolones—due to their well-documented impact on beneficial taxa such as *Bifidobacterium* and *Faecalibacterium*. Data were extracted on changes in major bacterial phyla, including Firmicutes, Bacteroidetes, Actinobacteria, and Proteobacteria, to evaluate shifts in community structure. This review also focuses on mechanisms of antibiotic resistance, including horizontal gene transfer and selective pressures favoring multidrug-resistant strains. Clinical implications such as increased risk of *Clostridioides difficile* infection and decreased short-chain fatty acid production were also analyzed. Furthermore, therapeutic strategies aimed at restoring microbial balance—such as fecal microbiota transplantation (FMT) and synbiotics—were evaluated for their effectiveness in enhancing resilience and colonization resistance. This review is based exclusively on secondary data; no new experimental research was conducted. All ethical standards for systematic literature reviews were followed. Based on the findings, we propose recommendations for antibiotic stewardship that emphasize the importance of targeted therapies to minimize dysbiosis and curb the spread of antibiotic resistance. The review adheres to the PRISMA Extension for Scoping Reviews (PRISMA-ScR) checklist [20], which was employed to ensure methodological rigor and transparent reporting. A visual summary of the study selection process is presented in Figure 1. Key reporting items addressed in this study are summarized in the PRISMA-ScR checklist (Table 1).

## 3. The Human Intestinal Microbiota: Composition and Function

### 3.1. Taxonomic and Functional Diversity

The human gastrointestinal microbiota is a vast, dynamic milieu of bacteria, archaea, fungi, viruses, and protozoa that collectively orchestrate nutrient processing, immunological modulation, and broader health outcomes [21]. Microbial density varies along the gastrointestinal tract: the small intestine has fewer microbes due to higher oxygen levels and faster transit, while the colon supports a richer and more stable community [22]. Predominant bacterial phyla include Firmicutes, Bacteroidetes, Actinobacteria, Proteobacteria, and Verrucomicrobia, each contributing to essential physiological functions. For instance, *Firmicutes* (*Clostridium*, *Faecalibacterium*) drive short-chain fatty acid (SCFA) production, while *Bacteroidetes* (*Bacteroides, Prevotella*) specialize in polysaccharide breakdown [23]. Additional minor constituents like *Akkermansia muciniphila* help sustain mucosal integrity [22,23]. Microbial diversity and composition shift over the lifespan and under external influences (e.g., diet, environment) [21].

These gut-resident microorganisms are pivotal in metabolic homeostasis, nutrient assimilation, and immune regulation. They ferment complex carbohydrates to produce SCFAs like acetate, propionate, and butyrate, fueling colonocytes and reducing inflammation [21]. In addition, they synthesize vitamins (e.g., vitamin B complex, vitamin K) and participate in amino acid metabolism and gut–brain axis signaling via microbial-derived neurotransmitter precursors [24]. Immune function is likewise modulated by the microbiota, as bacterial metabolites foster regulatory T-cell development, modulate antimicrobial peptide release, and interact with innate immune sensors to maintain immune balance [19]. A well-balanced and diverse gut microbiota thus not only deters pathogenic overgrowth (colonization resistance), but also preserves intestinal barrier function [25]. Conversely, dysbiosis—characterized by altered diversity and elevated pro-inflammatory taxa—underlies inflammatory bowel disease, obesity, type 2 diabetes, colorectal cancer, and certain neurological disorders [26]. Understanding the symbiotic relationships among microbes, dietary patterns, and host pathways is integral to developing targeted interventions (e.g., probiotics, prebiotics, FMT) aimed at re-establishing microbial homeostasis and preventing disease [27].

### 3.2. Role in Host Physiology and Immune System Modulation

Within the gastrointestinal tract, a highly tuned interplay between the host and its resident microbial communities underpins metabolic, immune, and overall physiologic stability [28,29]. SCFAs (notably butyrate, acetate, and propionate)—key microbial byproducts—modulate immune function by mitigating inflammation, shaping immune cell phenotypes, and reinforcing epithelial integrity [30,31]. Interactions between gut microbes and the immune system are primarily mediated through pattern recognition receptors (PRRs) on intestinal epithelial and antigen-presenting cells [24,32]. The innate immune system relies on epithelial barriers, antimicrobial peptides, and tissue-resident immune cells (e.g., macrophages, dendritic cells, innate lymphoid cells) to regulate microbial loads [33,34]. Dysbiosis, however, can breach these protective mechanisms, raising intestinal permeability and triggering chronic inflammation that propels inflammatory bowel disease, metabolic syndromes, and neurological dysfunctions via gut–brain axis pathways [35,36]. Adaptive immunity refines these responses; mucosa-associated lymphoid structures orchestrate immunoglobulin A (IgA) production and T-cell differentiation, fostering tolerance toward commensals or mounting immune responses to pathogens [37,38]. Meanwhile, specific SCFAs are particularly influential in encouraging regulatory T-cell (Treg) populations, diminishing inflammation, and supporting epithelial homeostasis [39]. Hence, the gut acts as an “immunological command center” moderated by microbial signals that drive context-dependent immune outcomes [40]. Advances in microbiome-based therapies—including probiotics, prebiotics, and FMT—offer new avenues for restoring microbial balance and bolstering immune resilience [41,42]. As ongoing research sheds light on these interactions, microbiome-targeted interventions promise potential for preventing and managing numerous immunologically mediated conditions.

### 3.3. Microbiota Stability and Resilience

Gut microbiota stability (i.e., sustained composition/function over time) and resilience (i.e., recovery after external stress) are critical for warding off dysbiosis-associated pathologies [43]. Microbial diversity, functional redundancy, and robust host–microbe signaling foster this equilibrium [44,45]. While broad-spectrum antibiotics strongly disrupt beneficial taxa, the microbiota retains an inherent capacity to rebound through functional redundancy, β-lactamase activity, and recolonization [46]. Diet also significantly modulates stability: high-fiber diets encourage SCFA-producing bacteria and fortify gut barrier integrity, whereas Western-style diets elevate disease risk by suppressing beneficial taxa [45,47]. Emerging interventions (e.g., probiotics, FMT, microbiome-driven dietary adjustments) underscore the growing impetus for “microbiome-first medicine” that prioritizes gut microbial health [47,48].

## 4. Impact of Antibiotics on the Intestinal Microbiota

Although antibiotics are vital for treating bacterial infections, they can dramatically disrupt the gut microbiota, a delicate ecosystem essential for immune maturation, metabolism, and overall health (Figure 2). Broad-spectrum agents often drive dysbiosis—resulting in a loss of diversity and the distortion of community composition—by depleting beneficial genera (*Bifidobacterium*, *Faecalibacterium*) and altering key phyla (Firmicutes, Bacteroidetes, Actinobacteria, Proteobacteria) [49,50]. Weakened colonization resistance paves the way for opportunistic pathogens (e.g., *C. difficile*, *Salmonella typhimurium*) [51,52], while reduced SCFA production further undercuts gut homeostasis [6,49]. Antibiotics can also accelerate antibiotic resistance via horizontal gene transfer [6]. Recovery from these perturbations hinges on antibiotic spectrum, dose, and duration, potentially taking months or years in some cases [49]. Early-life disruptions may incur lasting immunological, metabolic, and cognitive repercussions. Probiotic or synbiotic supplementation and selective antibiotic regimens can mitigate such adverse outcomes, highlighting antibiotic stewardship’s criticality [50,53]. See Table 2 for an overview of microbiome-based therapeutic approaches.

## 5. Mechanistic and Technological Insights into Dysbiosis and Antimicrobial Resistance

Dysbiosis—defined by the loss of beneficial microbes, the overgrowth of pathogens, and reduced microbial diversity—results from multifactorial disturbances, such as antibiotic exposure, immune dysregulation, metabolic imbalance, and environmental stressors [54,55]. These changes not only disrupt gut homeostasis, but also promote the emergence and spread of antimicrobial resistance.

### 5.1. Disruption of Microbial Homeostasis and Pathophysiological Consequences

A stable gut ecosystem is primarily sustained by dominant phyla such as Firmicutes and Bacteroidetes, which modulate immune function, produce short-chain fatty acids (SCFAs), and maintain mucosal integrity [56]. Disruptive factors—including antibiotic use, low-fiber diets, and inflammation—can destabilize this balance, favoring the expansion of opportunistic pathogens like *Enterobacteriaceae* [57]. Dysbiosis increases intestinal permeability, facilitating the translocation of microbial components (e.g., lipopolysaccharides) into systemic circulation and triggering inflammation [55,58]. Concurrently, the loss of SCFA-producing bacteria impairs the differentiation of regulatory T cells (Tregs), intensifying inflammatory responses [56].

### 5.2. Epithelial Crosstalk and Systemic Implications

The interaction between gut epithelial cells and the microbiota plays a critical role in immune education and barrier integrity. Specialized epithelial cells—such as tuft and Paneth cells—engage in bidirectional communication with commensals to uphold intestinal architecture and immune surveillance [57,58]. The disruption of this signaling axis not only contributes to gastrointestinal disorders, but also exerts systemic effects, including immune dysfunction and neurological disturbances via the gut–immune and gut–brain axes [55,58,59].

### 5.3. Dysbiosis and Antibiotic Resistance

Antibiotic-induced dysbiosis accelerates the horizontal transfer of antimicrobial resistance genes (ARGs) within the gut microbiota, compromising colonization resistance and enabling the proliferation of multidrug-resistant organisms [6,50,60]. The depletion of key taxa—such as *Faecalibacteriu prausnitzii* and *Bifidobacterium* spp.—further increases susceptibility to resistant strains, including extended-spectrum β-lactamase (ESBL)-producing *Enterobacteriaceae* [61,62]. Emerging interventions like fecal microbiota transplantation (FMT), targeted probiotics, and postbiotics aim to counter these risks while preserving the integrity of the broader microbial ecosystem [4,50].

### 5.4. Integrative Multi-Omics and Computational Approaches

To unravel the complexity of antibiotic-driven dysbiosis, cutting-edge multi-omics platforms—such as metagenomics, metatranscriptomics, metabolomics, and metaproteomics—are increasingly employed [63,64]. These technologies allow the high-resolution profiling of microbiota composition, functional pathways, and resilience dynamics [65,66]. For instance, Zhernakova et al. [67] used multi-omics to identify the antibiotic-associated depletion of *Faecalibacterium* and reduced butyrate synthesis—key indicators of microbiota stability. Similarly, Manor et al. [68] applied genome-scale metabolic models to predict SCFA production under various antibiotic regimens, informing the design of microbiota-preserving therapies.

Machine learning further supports these efforts by detecting critical “tipping points” within microbial networks that predict dysbiosis onset. Shi et al. [69], for example, demonstrated how *Lactobacillus*-based interventions can modulate immune responses and restore microbial balance following β-lactam-induced dysbiosis.

### 5.5. Biomarker Discovery and Immune Network Mapping

Targeted metabolomics has identified bile acid derivatives and aromatic amino acid metabolites as early biomarkers of dysbiosis and ARG propagation [70]. These indicators are now incorporated into experimental diagnostic platforms to guide therapeutic decisions [71]. Simultaneously, systems biology approaches have enabled the detailed mapping of immune–microbial interactions, revealing key regulatory nodes such as IL-22-mediated epithelial repair [72,73].

### 5.6. Translational Therapeutics and Clinical Applications

Innovative microbiota-based therapies—shaped by these mechanistic insights—include synthetic microbial consortia like RePOOPulate, designed to restore fermentation capacity and mucosal immunity in *Clostridioides difficile* infections [74], and bacteriophage cocktails that selectively target resistant pathogens while sparing commensals [75]. As computational tools evolve, these precision-guided approaches are transforming microbiome therapeutics, enabling individualized antibiotic stewardship and limiting resistance development [68,76].

### 5.7. Clinical Restoration Strategies and Regulatory Challenges

Microbiome-targeted interventions—including probiotics, synbiotics, and FMT—have shown promise in treating *C. difficile* infection, inflammatory bowel disease, and metabolic syndromes [69,70,71,72,73,74,75,76,77,78,79,80]. Nevertheless, concerns about donor variability, safety, and regulatory consistency—especially for FMT and live biotherapeutic products—persist [81,82]. Standardization in manufacturing, the development of defined microbial formulations, and clear regulatory pathways will be crucial for ensuring safe and scalable implementation.

## 6. Cutting-Edge Mechanistic Insights and Innovative Therapeutic Frontiers

Recent advances in microbiome science have catalyzed a shift from empirical treatments to precision interventions, grounded in the mechanistic understanding of host–microbiota dynamics in the context of antibiotic perturbation [83,84]. This section highlights breakthrough strategies that leverage systems-level insights for targeted microbiota restoration.

### 6.1. Systems Biology and Network Analysis of Host–Microbiota Interactions

Contemporary systems biology enables the modeling of complex microbial ecosystems with unprecedented resolution. For example, studies by Zmora et al. [85] have integrated single-cell transcriptomics with spatial omics to reveal how antibiotics disrupt mucosal-layer colonization by key commensals, leading to impaired IL-10 signaling and increased susceptibility to inflammation. These network analyses have elucidated central regulatory nodes, such as *Akkermansia muciniphila*-driven mucin degradation pathways, which can be therapeutically modulated to restore epithelial integrity [86].

### 6.2. Precision Microbiome Therapeutics: Next-Generation Strategies

Novel therapeutic strategies now focus on restoring specific microbiome functions rather than broad recolonization. For instance, Petrof et al. [87] demonstrated the efficacy of a defined microbial ecosystem (RePOOPulate) in resolving recurrent *C. difficile* infection, offering a safer alternative to fecal microbiota transplantation. Similarly, bacteriophage therapy has shown promise in selectively depleting pathogenic *Escherichia coli* while preserving beneficial taxa, as reported in mouse models by Li et al. [88]. Postbiotics—non-viable microbial products such as butyrate-rich vesicles or purified microbial enzymes—are being developed to restore metabolic signaling and epithelial function without introducing live microbes, reducing the risk of unintended colonization or immune reactions [89].

### 6.3. Translational Implications and Future Directions

The convergence of high-resolution omics, predictive analytics, and user-centered bioinformatics tools paves the way for real-time, personalized microbiome monitoring. Platforms like MICROSCOPE and the Human Microbiome Cloud (HMC) now offer clinicians dashboards to track microbial shifts and receive evidence-based recommendations for intervention [90]. Pilot studies integrating these tools into antimicrobial stewardship protocols have reported a reduced incidence of microbiota-associated adverse effects and lower rates of multidrug-resistant organism colonization [91]. Looking ahead, clinical protocols may incorporate routine microbiome profiling to inform dynamic treatment adjustments—such as the pre-emptive administration of narrow-spectrum antibiotics, adjunctive synbiotics, or phage cocktails—to harmonize infection control with the preservation of host–microbial homeostasis.

## 7. Knowledge Gaps and Future Directions

Despite significant advances in microbiome science, critical knowledge gaps remain that limit the precision, scalability, and safe clinical deployment of microbiota-targeted interventions. A key challenge is the substantial inter-individual variability in microbiota recovery following antibiotic exposure, influenced by host genetics, immune status, lifestyle, and antibiotic pharmacodynamics [92]. While some individuals achieve microbial restoration within weeks, others exhibit prolonged dysbiosis, increasing susceptibility to chronic inflammation and disease [46,54]. This variability is compounded by the absence of validated biomarkers of microbiome resilience, which hinders risk stratification and the development of personalized therapies [47,70]. Importantly, host-related factors such as age, diet, and immune competence have a significant impact on both the extent of dysbiosis and the pace of microbiota recovery. For instance, elderly individuals and immunocompromised patients often experience more profound and prolonged microbiota disruptions [93,94].

Diet, particularly fiber intake and dietary diversity, also plays a key role in shaping the trajectory of microbial reconstitution post-antibiotics [95,96]. These dimensions remain underexplored in current clinical frameworks and deserve greater emphasis in future study designs.

Another critical issue is the methodological heterogeneity in studies investigating antibiotic-induced microbiome alterations. While shotgun metagenomics enables the high-resolution taxonomic and functional profiling of microbial communities, it lacks the ability to capture viable and metabolically active strains. Conversely, culture-based methods, although limited in scope, offer functional insights and strain-level isolation for downstream applications [97,98]. A systematic comparison of these approaches—and potentially integrative methodologies—is needed to standardize outcome measures and facilitate cross-study comparison.

The lack of standardization in fecal microbiota transplantation (FMT) also presents a major translational barrier. Although FMT remains highly effective for recurrent *Clostridioides difficile* infections [52], it carries risks related to safety, reproducibility, and the transmission of antimicrobial resistance genes or undesirable metabolic traits [67,88]. Consequently, attention is shifting toward defined microbial consortia, engineered live biotherapeutics, and postbiotics, which offer improved control, reproducibility, and regulatory compliance [79,90].

Moreover, although the gut–immune and gut–brain axes are increasingly recognized as critical mediators of systemic responses to dysbiosis [10,11], the mechanistic links between microbial disruption and downstream host dysfunction remain poorly defined. For example, the antibiotic-induced depletion of short-chain fatty acid (SCFA)-producing taxa such as *Faecalibacterium prausnitzii* may impair immune homeostasis and contribute to chronic inflammatory states [76].

To address these gaps and enhance the translational potential of microbiome-based interventions, future research should prioritize the following:Longitudinal, multi-omics studies to define microbial and host signatures of resilience and vulnerability [70,74];Standardized FMT protocols, including donor screening, microbial quality control, and long-term safety monitoring [52];The clinical validation of next-generation microbiome-based therapeutics, with clearly defined safety and efficacy profiles [79,88];Mechanistic studies integrating host transcriptomics, immunophenotyping, and metabolomics to elucidate causal pathways linking dysbiosis to disease [74,76].

## 8. Future Perspectives

Looking ahead, precision microbiome medicine is expected to transform clinical approaches to managing antibiotic-induced dysbiosis and related disorders. Advances in metagenomics, metabolomics, and machine learning are enabling the development of real-time diagnostics that can guide individualized treatment decisions and minimize collateral damage to the gut microbiota [93,94,95]. Artificial intelligence-driven analytics can now predict microbiome vulnerability and recovery potential, thereby supporting dynamic antibiotic stewardship strategies [91,96]. For example, narrow-spectrum antibiotics and β-lactamase inhibitors are being evaluated as microbiota-sparing alternatives to broad-spectrum regimens [97]. In parallel, personalized microbiota restoration therapies, such as targeted probiotics, synthetic microbial consortia, and postbiotic compounds, are under active development [90,98].

Furthermore, personalized dietary interventions—such as prebiotic-enriched or fiber-rich diets—may be employed to promote microbial resilience and maintain homeostasis during and after antibiotic therapy [49,99]. These strategies align with a broader shift toward “microbiome-first medicine”, where therapeutic decisions account for host–microbiota dynamics [48]. To ensure successful clinical translation, healthcare systems should undertake the following:Invest in infrastructure for point-of-care microbiome analysis [100];Develop regulatory frameworks for the approval and monitoring of microbiota-based therapeutics [98];Promote interdisciplinary collaboration among clinicians, microbiologists, computational biologists, and regulatory bodies [81,92].

This paradigm shift has the potential to reduce antibiotic-associated complications, preserve microbial diversity, and enhance patient outcomes through sustainable, individualized microbiome management.

## 9. Conclusions

Antibiotic therapy remains essential for the management of bacterial infections, yet its unintended disruption of the gut microbiota presents significant clinical challenges [101]. Broad-spectrum agents often compromise beneficial taxa, facilitate pathogen overgrowth, and impair host–microbiome homeostasis [102]. The heterogeneity of microbiome recovery highlights the need for precision approaches in mitigating antibiotic-induced dysbiosis, as outcomes depend on antibiotic class, dosage, duration, and host-specific factors [103]. Emerging strategies—including targeted probiotics, microbiome-based therapeutics, and integrative multi-omics tools—offer new opportunities to preserve or restore microbial balance [104]. Moving forward, personalized antibiotic regimens supported by real-time microbiome monitoring and rational microbial restoration will be critical to improving patient outcomes and promoting sustainable antibiotic stewardship [105].

## Figures and Tables

**Figure 1 antibiotics-14-00371-f001:**
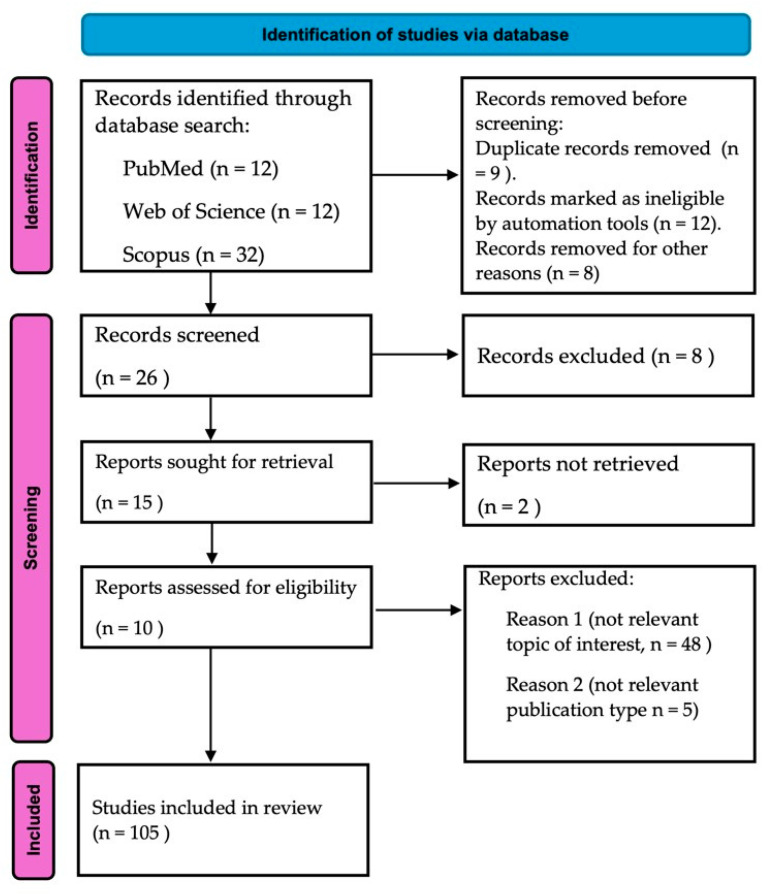
Study flowchart following the PRISMA Extension for Scoping Reviews (PRISMA-ScR) guidelines [20]. The diagram illustrates the sequential phases of identification, screening, eligibility assessment, and the final inclusion of studies. It provides a clear overview of the number of records retrieved, excluded, and ultimately included in the review.

**Figure 2 antibiotics-14-00371-f002:**
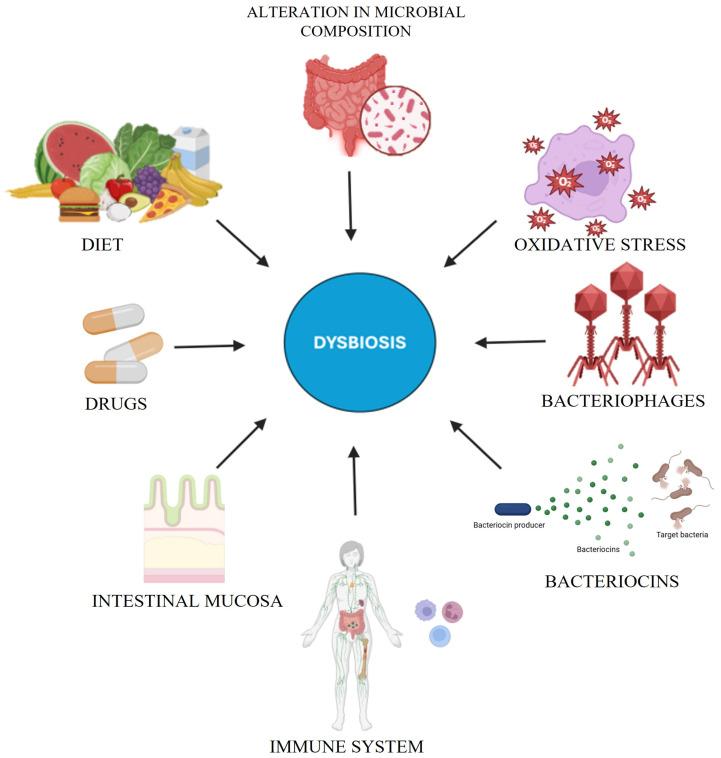
This diagram depicts the multifactorial contributors to dysbiosis, a disruption of microbial homeostasis. Key factors include diet, drugs, the intestinal mucosa, and the immune system, which regulate microbial stability. Oxidative stress, bacteriophages, and bacteriocins further modulate microbial composition, while microbial alterations can sustain dysbiosis, compromising gut and systemic homeostasis.

**Table 1 antibiotics-14-00371-t001:** PRISMA Extension for Scoping Reviews (PRISMA-ScR) checklist [20]. This table summarizes the key reporting items, their corresponding descriptions, and the extent to which each was addressed in the present study. The checklist supports methodological rigor and ensures comprehensive reporting throughout the review process.

Section	Requirement	Present in Manuscript
Title	Identify as a systematic review, meta-analysis, or both.	Yes
Abstract	Structured summary including background, methods, results, and conclusion.	Yes
Rationale	Describe the rationale for the review.	Yes
Objectives	Provide an explicit statement of the objectives.	No
Eligibility Criteria	Specify study eligibility criteria.	No
Information Sources	Specify all information sources (e.g., databases, dates).	Yes
Search Strategy	Present full search strategy for at least one database.	No
Selection Process	Describe selection process (screening, eligibility, inclusion).	No
Data Collection Process	Describe methods of data extraction and management.	No
Data Items	List all variables and outcomes collected.	No
Study Risk of Bias Assessment	Describe methods for assessing risk of bias.	No
Effect Measures	Describe all methods for effect estimation.	No
Synthesis Methods	Specify how results were synthesized.	No
Certainty Assessment	Assess certainty in evidence (e.g., GRADE framework).	No
Results	Report number of included studies and characteristics.	Yes
Discussion	Discuss results in context of limitations and strengths.	No
Funding	Disclose funding sources and conflicts of interest.	Yes

**Table 2 antibiotics-14-00371-t002:** Advantages, disadvantages, and future directions of microbiome-based therapeutics. A concise overview of microbiome-based therapies, emphasizing their benefits, limitations, and future prospects for enhanced precision and clinical effectiveness.

Therapeutic Approach	Mechanism of Action/Examples	Advantages	Disadvantages	Regulatory Status and Clinical Trials	References
Probiotics	Live microorganisms conferring health benefits. Common strains: *Lactobacillus rhamnosus* GG, *Bifidobacterium longum*, *Saccharomyces boulardii*.	Support microbiota restoration, enhance immune tolerance, and modulate inflammation.	Strain-specific efficacy; variability in colonization and host response.	Approved as dietary supplements in most countries; multiple RCTs for IBS, CDI, and AAD.	[50,53]
Prebiotics	Non-digestible fibers that selectively stimulate growth/activity of beneficial bacteria. E.g., inulin, FOS, and GOS.	Improve SCFA production, enhance barrier function, and modulate glucose/lipid metabolism.	Efficacy depends on host microbiota baseline; inconsistent clinical outcomes.	GRAS status (Generally Recognized as Safe); RCTs ongoing for metabolic syndrome and IBD.	[49,50]
Synbiotics	Combination of probiotics + prebiotics designed for synergistic effect. E.g., *Lactobacillus plantarum* + inulin.	Enhanced colonization and metabolic impact; better microbiota resilience.	Formulation complexity; limited standardization across products.	Studied in RCTs for CDI, NEC in neonates, and hepatic encephalopathy; no unified regulatory classification.	[6]
Fecal Microbiota Transplantation (FMT)	Transplantation of processed stool from healthy donors.	Restores entire microbial ecosystem; highly effective against recurrent C. difficile.	Risk of pathogen transfer; regulatory and ethical concerns; donor variability.	FDA allows use under IND applications; multiple RCTs. Approved in Canada for CDI.	[5,51]
Postbiotics	Bioactive metabolites/products from microbes (e.g., butyrate, bacterial cell wall fragments, extracellular vesicles).	No live bacteria—lower infection risk; modulate immune and metabolic pathways.	Heterogeneity in composition; lack of standardized production and dosage.	Preclinical and early-phase human trials; regulatory definitions evolving.	[53]
Engineered Microbiota	Genetically modified or synthetic consortia tailored for therapeutic function. E.g., RePOOPulate, engineered E. coli Nissle.	Precision targeting of functions (e.g., SCFA production, ARG suppression).	Complex safety assessments; unknown ecological impacts; high regulatory hurdles.	Experimental stage; early-phase trials in CDI, IBD, and metabolic disorders.	[6]

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
