# Peer review of "The Impact of Antibiotic Therapy on Intestinal Microbiota: Dysbiosis, Antibiotic Resistance, and Restoration Strategies"

_antibiotics, 2025, doi:10.3390/antibiotics14040371_

Round 1
Reviewer 1 Report
Comments and Suggestions for Authors
This comprehensive review article provides an insightful analysis of the impact of antibiotic therapy on the human intestinal microbiota, detailing mechanisms of dysbiosis, resistance, and various restoration strategies. The review is well-structured, citing a broad range of recent literature to support its arguments. The integration of multi-omics and in silico approaches highlights the cutting-edge nature of the research, positioning the article as a valuable contribution to the field. However, there are a few areas that could benefit from clarification and minor revisions to enhance the quality of this manuscript.
- To make the title more clear, it is recommended to change “Resistance” to “Antibiotic Resistance”
- Check the consistency of specialized vocabulary, e.g Line 25 ”Precision microbiome therapies” and Line 311 “Precision Microbiome Therapeutics”.
- Avoid the passive voice where possible, e.g., change "Antibiotics are vital in treating bacterial infections" (line 43) to "Antibiotics play a vital role in treating bacterial infections."
- Figure 1 is dull and simple, and I recommend redrawing it.
- In section 8. Future Perspectives, the authors might expand on lines 333-339 to provide more concrete examples and recommendations.
Author Response
1_To make the title more clear, it is recommended to change “Resistance” to “Antibiotic Resistance”
Answer: We agree with this comment. Therefore, to clarify the title, we have changed “Resistance” to “Antibiotic Resistance.”
2_Check the consistency of specialized vocabulary, e.g Line 25 “Precision microbiome therapies” and Line 311 “Precision Microbiome Therapeutics”.
Answer: "Precision Microbiome Therapeutics" is the more appropriate term
3_Avoid the passive voice where possible, e.g., change "Antibiotics are vital in treating bacterial infections" (line 43) to "Antibiotics play a vital role in treating bacterial infections."
Answer: On page 2, line 43, the sentence "Antibiotics are vital in treating bacterial infections" has been replaced with "Antibiotics play a vital role in treating bacterial infections." We sincerely thank Reviewer 1 for this attention to detail.
4_Figure 1 is dull and simple, and I recommend redrawing it.
Answer: Figure 1 has been replaced with a more relevant and, hopefully, more engaging image. However, the insertion of the new figure has altered the line numbering in the subsequent text. The authors thank Reviewer 1 for the valuable suggestion.
5_In section 8. Future Perspectives, the authors might expand on lines 333-339 to provide more concrete examples and recommendations.
Answer: In section 8, Future Perspectives, the authors have expanded on lines 333-339 to provide more concrete examples and recommendations. We sincerely thank Reviewer 1 for this attention to detail.
Reviewer 2 Report
Comments and Suggestions for Authors
The impact of antibiotics on gut microbiota is significant, making it essential to provide a comprehensive review of current research in this field. In this article, Cusumano et al. effectively summarize the causes of antibiotic-induced dysbiosis and propose potential restoration strategies. The review is well-structured and informative, but there is still space for improvement:
- Lack of Specific Examples:
Some descriptions in the manuscript are too vague and lack sufficient detail. For instance, in Sections 6 and 7, the authors introduced several innovative strategies for microbiota restoration but only provided citations without further elaboration. Including a representative example from these references to illustrate each strategy would greatly enhance clarity and help readers better understand the practical implications of these advancements. - Additional Figure for Sections 5-7:
The existing tables and figures are helpful in summarizing key points from earlier sections. However, if the authors could provide an additional figure or schematic that consolidates the strategies discussed in Sections 5-7, it would make the information more accessible and easier for readers to grasp the core concepts.
Author Response
Lack of Specific Examples:
Some descriptions in the manuscript are too vague and lack sufficient detail. For instance, in Sections 6 and 7, the authors introduced several innovative strategies for microbiota restoration but only provided citations without further elaboration. Including a representative example from these references to illustrate each strategy would greatly enhance clarity and help readers better understand the practical implications of these advancements
Answer: In Sections 6 and 7, the authors have added further insights by including representative examples. Additionally, concise summary tables outlining tools/approaches, functions, and clinical or research applications have been included, as shown in Tables 2 and 3. Thank you for pointing this out.
Additional Figure for Sections 5-7:
The existing tables and figures are helpful in summarizing key points from earlier sections. However, if the authors could provide an additional figure or schematic that consolidates the strategies discussed in Sections 5-7, it would make the information more accessible and easier for readers to grasp the core concepts.
Answer: In Section 5, Figure 1 has been replaced with a more meaningful image, while in Sections 5 and 6, two tables have been added to help summarize key points from the earlier sections.
Reviewer 3 Report
Comments and Suggestions for Authors
This manuscript provides a comprehensive and well-researched review of how antibiotic therapy affects intestinal microbiota, focusing on dysbiosis, resistance mechanisms, and microbiota restoration strategies. Also, the manuscript integrates current literature with emerging trends in microbiome-based therapies, multi-omics, and systems biology. It is timely and relevant given the global concern over antibiotic resistance and the increasing attention toward gut microbiota's role in health.
However, several areas require improvement in terms of critical analysis, organization, and clarity to maximize the manuscript’s impact. Below are my specific comments for improvements.
Major concerns:
- Novel perspective
While it covers a broad range of topics, it lacks a distinct novel perspective or a strong argument about how this review advances the field beyond existing literature. Consider introducing a novel conceptual diagram or framework integrating dysbiosis, resistance development, and microbiota-targeted interventions. The authors may consider adding a section “Knowledge Gaps & Future Directions” to summarize areas that remain unresolved, such as how to reduce variability in microbiome recovery after antibiotic exposure and minimize any off-targets effects for FMT.
- Table
Table 1 – may need add more in-depth information. The table is useful but a little too simplified. Consider expanding it with specific examples for each therapeutic (e.g., probiotic strain names), also current regulatory status and whether the approaches are currently in clinical trials (Randomized Controlled Trials or Observational Studies).
- Scientific critical analysis
- For example, comparison of different methodologies used to study antibiotic-induced microbiota changes (e.g., metagenomics vs. culture-based methods).
- How microbiome alterations vary depending on age, diet, and immune status (more emphasis needed on host factors influencing dysbiosis severity).
Minor concerns:
- Organization
Consider combining Sections 5 and 6, as both discuss mechanisms of dysbiosis and antibiotic resistance. These concepts are interrelated and merging them would reduce repetition.
- Methodology section
Consider including:
- Number of studies reviewed,
- Criteria for inclusion/exclusion (e.g., animal vs. human studies),
- Formatting
- Some references in the text (e.g., “(52,53]”) need to be more consistent.
- Break up long sentences, especially in sections discussing mechanisms of dysbiosis.
- Ensure uniform reference style per journal guidelines.
- SCFAs, FMT, ENS, etc., are defined in the beginning but not consistently throughout. Consider including an abbreviation table.
- Some references in the text (e.g., “(52,53]”) need to be more consistent.
- Break up long sentences, especially in sections discussing mechanisms of dysbiosis.
Author Response
please see our reply to your comments in the attachment

Reviewer 4 Report
Comments and Suggestions for Authors
Dear Author's
Throughout each and every methods, all looks like introduction of the technique pertaining to the topic. It looks like lack of in depth review. All data must be very specific and may not be superficial. For example, If you take conclusion part, there you have discussed few things which looks good. But it is conclusion section not a discussion. Hence, either you can keep discussion section separately or else discuss in details with adequate proof of data under each topic. Also, do conclude very short and crisp.
Author Response
Dear Author's
Throughout each and every methods, all looks like introduction of the technique pertaining to the topic. It looks like lack of in depth review. All data must be very specific and may not be superficial.
For example, If you take conclusion part, there you have discussed few things which looks good. But it is conclusion section not a discussion. Hence, either you can keep discussion section separately or else discuss in details with adequate proof of data under each topic. Also, do conclude very short and crisp.
Answer: Several topics were explored in greater depth in Sections 6 and 7, supported by adequate data evidence.
The authors sincerely thank Reviewer 4 for the valuable suggestions, which have significantly contributed to improving the quality of the manuscript
Round 2
Reviewer 4 Report
Comments and Suggestions for Authors
Dear Authors,
Revised manuscript looks improved as suggested.